# Rurality representation and changes in rural tourism destination

**Xueting Hong**[1], **Zhenfang Huang**[2], **Xinyi Li**[1], **Yechen Zhang**[3]*

1 School of Geography and Tourism, Anhui Normal University, Wuhu, China, 2 School of Geographical Science, Nanjing Normal University, Nanjing, China, 3 College of Tourism and Exhibition, Hefei University, Hefei, China

* zhangyc@hfuu.edu.cn

## Abstract

Representation of rurality plays a crucial role in developing rural tourism. However, the connotations of rural destinations and how such representations shift across different social contexts remains unclear. This study aims to examine the rurality of, and changes in, rural tourism destinations through the lens of social representation theory. Semi-structured interviews were conducted with 50 residents and 17 tourists from rural destinations in Nanjing, China. We identified three core dimensions of rurality representation in these destinations: material, social, and cultural. From these, six primary categories emerge: the natural environment, settlement architecture, production and daily life, interpersonal relationships, folk customs and festivals, and spiritual orientation. The findings indicate that the representation of rurality in rural tourism destinations has evolved from an agriculture-centred to a tourism-centred focus. We confirm that rurality is socially constructed and varies according to social background. Rural tourism development should consider the nuanced connotations of rurality and adopt a dynamic perspective to foster high-quality growth in the sector.

## 1. Introduction

The crisis in rural villages has been driven by industrialisation, which has prompted the concentration of populations in major cities. Rural villages face pressing challenges, including population decline, aging demographics, economic downturns, and the loss of regional characteristics; consequently, rurality can be gradually eroded or even disappear [1–3]. Many countries have recognised these unique challenges and have implemented various development and policy initiatives. China addresses the 'Three Rural Issues', encompassing agriculture, rural development, and the welfare of farmers [4,5]. Policies, such as 'The Beautiful Countryside' and 'Rural Revitalization' [6], aim to preserve traditional elements of rural life while improving living environments. Rural space embodies the essence of civilisation and represents

**Data availability statement:** All relevant data are within the manuscript and its Supporting Information files.

**Funding:** This work was supported by the National Natural Science Foundation of China (Grant No. 42301272) and the Humanities and Social Science Foundation of the Ministry of Education (Grant No. 22YJC790041). The funders had no role in study design, data collection and analysis, decision to publish, or preparation of the manuscript. There was no additional external funding received for this study.

**Competing interests:** The authors declare they have no conflicts of interest.

a valuable resource for tourism. As an organic whole, rurality is a complex system containing numerous functional dimensions, including ecology, economy, society, culture, and aesthetics. Understanding rurality requires considering its ontology and the dialectical relationships within rural spaces.

In industrialised societies, people increasingly seek to escape urban life and pursue long-term experiences in rural environments [7,8]. Rural tourism is an important way for urban escape, making rural revitalisation a central concern. Rurality has gained widespread recognition in research and practice as a primary attraction for rural tourism [9]. However, the development of rural tourism urbanises rural areas, commercialises cultural practices, and 'stages' cultural products, potentially degrading rural spaces and undermining sustainable tourism development. Therefore, expressing, maintaining, and enhancing the socially constructed representations of rurality is crucial, as preserving rural character and landscapes is a fundamental goal of rural tourism development.

Social representation theory posits that knowledge, beliefs, and common sense are symbolic forms shared socially through communication [10]. This theory has been widely applied in environmental studies, gender studies, image research, rural studies, and tourism, with particular focus on cognitive differences between groups and the processes through which social representations are formed.

The representation of rurality has been explored across countries, regions, and social groups [11]. In tourism destinations, rurality is a social representation of rural space, showing natural and cultural characteristics of regional group identity [12]. From the perspective of social representation theory, rurality is influenced by the social environment [13]. Representations of rurality is based on external reality but may deconstruct and create reality through the interpretation of discourse [11]. Its core meaning changes alongside social values, exhibiting the characteristics of change and replacement in different times. While existing literature recognises the socially constructed and dynamic nature of rurality perception, it remains largely confined to static depictions or unidimensional analyses [14]. Moreover, how rurality perception is multi-dimensionally constituted and how its representations evolve in rural tourism contexts remain important yet underexplored questions. Therefore, a comprehensive empirical understanding of rurality's multidimensional essence and evolutionary in the context of rural tourism destinations remains an important scholarly frontier. This theoretical gap underscores the need for a systematic empirical investigation to comprehensively deconstruct the representations of rurality and their transformations under tourism influences.

Guided by social representation theory, this study selected three representative rural tourism destinations to investigate the core content of rurality representations. Through in-depth interviews with residents and tourists, this study analysed the specific connotations of typical rurality representations. The representational system of rurality, which serves as a fundamental attraction for rural tourism, is the focus [15]. The representations of rurality in tourism destinations are diverse, highlighting the necessity of systematically identifying their sub-dimensions and connotations. This study examines two periods – before and after the development of tourism – to

assess how representations of rurality are altered by social change. Given the socially constructed nature of rurality, the study also considers differences between residents' and tourists' perceptions of rurality in tourism destinations. An accurate understanding of rurality representations can enhance destination competitiveness. Discussions on these representations contribute to a scientific understanding of rurality, inform policymakers in updating rural development strategies, provide guidance for constructing rural spaces, and enhance the core appeal of rural tourism destinations.

## 2. Literature review

### 2.1 Rurality

Rurality is a key concept that evolved from the term 'rural' to capture the essence of rural areas, and refers to the character of the rural as distinct from the urban [16]. Defining rurality is complex because rural areas exhibit diverse characteristics and change dynamically; consequently, a rural area may gradually transform into an urban area. Scholars have approached rurality from the perspectives of space, territory, and constructivism [17–20].The 'space' perspective views rural areas as the product of interactions between agglomeration and dispersion forces, forming systematic regional spaces encompassing production and housing. The 'territory' perspective emphasises the internal structure of villages, highlighting differences through concepts such as local production systems, industrial zones, and environmental innovations. The 'constructivist' perspective considers rural space as emerging from the interactions of social groups within a place.

Research on rurality has followed two main trajectories. The first employs a rurality index to quantitatively assess rural areas, anchoring the concept in observable and measurable physical attributes. The second prioritises sociocultural dimensions in conceptualising rurality [15]. Cloke [16] initially introduced the concept of rurality. He concentrated on its core essence and attempted to delineate and comprehend rural spaces through the construction of a rurality index. This framework emphasises the objective, materialist dimension of rurality. Subsequent studies have similarly sought to define rural areas by developing measurement indicators – often based on population density or proximity to metropolitan centres – to distinguish rural from urban locales [16,21–25].

Other studies have recognised that the characterisation of rurality is inherently diverse and adopted cultural definitions, influenced by group-specific and region-specific attributes from subjective viewpoints. Halfacree [18] contends that the definition of 'rural' depends on the social practice of rurality. Woods [26] advocates for a research focus that integrates material elements with spiritual considerations. As a product of social construction, rurality is fundamentally a subjective perception and evolves alongside shifting cultural sensibilities and historical contexts [27,26]. Consequently, in tourism settings where rurality is perceived by visitors, its definition should be derived from public perception rather than relying solely on objective indicators [28].

The descriptive definition is criticised for lacking dynamic and cultural dimensions, while sociocultural definitions are sometimes perceived as overly subjective. Research on rurality has deepened understanding of its dual nature: it is both a subjective perception and a socially constructed phenomenon. In rural studies, rurality is often quantified using objective indicators such as population density and economic structure. However, in tourism contexts, visitors' subjective perceptions of rurality are critical [29].This distinction highlights that while rurality may have an objective foundation, its meaning and significance are continually shaped through social interpretation and interaction [30].

As rural characteristics evolve with societal development, the essence of rurality is continually redefined. Its meaning emerges through social representation, a process in which shared understanding is negotiated among members of society. Thus, rurality functions as a dynamic system of shared meanings, continuously produced and reproduced through social interaction, discursive practice, and lived experience within specific sociohistorical contexts [31]. Although the nature of rurality is not debated widely, differences remain in its operational definitions. This study adopts a qualitative approach, treating rurality as a subjective perception – a methodological innovation in the field of rural tourism.

## 2.2 Social representation theory

Social psychologist Moscovici [10] posited the social representation theory, which emphasised that social knowledge is a process about how shared representation consensus is derived from social interactions [32]. This theory begins from two basic principles. The first considers the relationship between individuals' subjective and the objective world. It believes that the individual's subjective world co-constructs social reality. Second, the social representation theory uses discourse to determine the consensus formed. It categorises social representation into: hegemonic, emancipated, and polemical representations, based on the relationship between the community members [33]. Hegemonic representations are unitary and coercive, emancipated are subgroups that interact with each other, and polemical are characteristic of social conflict and controversy [34]. Social representations have three directions: anchoring, language and speech, and structural approach [33,35]. Anchoring emphasises the process from unknown to received, and language determines meaning in the environment. The structural method is the relationship between elements rather than element content and plays an important role. Social reality is pluralistic and diverse because it is constructed through conversations and discourses in daily work and different social groups. According to social representation theory, as the objects of representation change, so does people's collective consensus about that representation. Social representation is an important feature of social groups. Members of these groups share a set of beliefs that can essentially define an individual's social identity. Social representations are not static and can change over time between and within cultures through the interactions of individuals with other groups [36].

Bauer and Gaskell [37] developed a paradigm for research about social representation. In the context of this theory, a distinction should be made between social milieus and taxonomic clusters, as content is functional within particular social milieus and communities. In the 'Toblerone' model (Fig 1), the triangle represents the contents of social representation, which can be generally understood as social consensus. Social representation results from interaction between certain variables and objects and between subjects. At time t, interaction between subject and object forms one representation; at time t-1, which is the past, another form of social representation is formed. People form different understandings of the same subject at different times, and different representations are accepted to varying degrees at different times. According

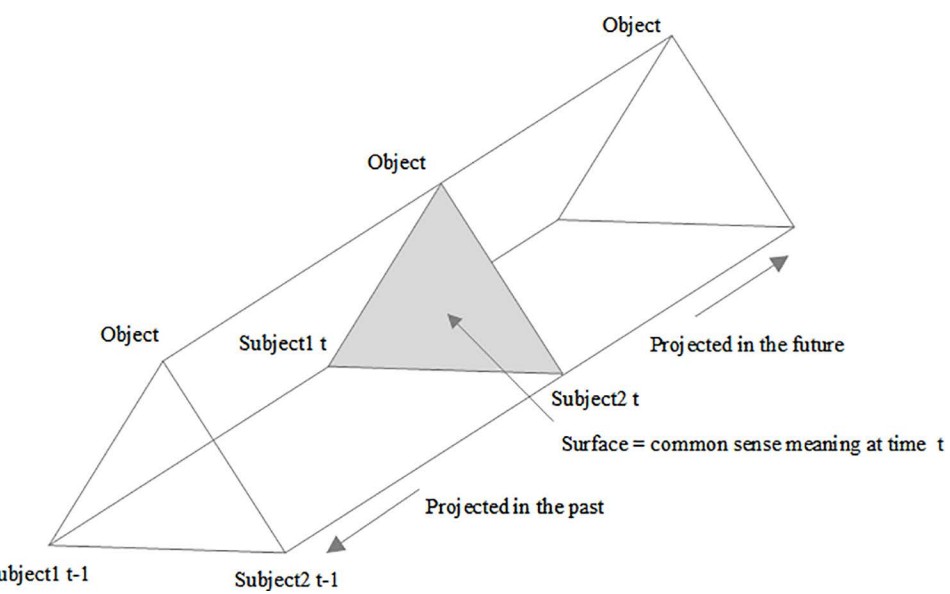

**Fig 1. Toblerone model of social representation.**

to Moscovici [10], consensus on the representation of rurality changed over time with changes in social background. Social representation theory is useful for the tourism context [38].

## 2.3 Social representations in rural tourism research

Based on social representation theory, research on rurality has focused on themes such as the rural idyll, representations of rurality, social groups (e.g., gender, children), discourse, community, and rights and interests. It further broadened to understand rural space as a subjective social construct, rather than an objective material reality. From the perspective of social representation theory, the connotation of rurality reflects the collective understanding of rural characteristics, shaped by social practices and values of the time, and evolving with changes in social context and the environment. As rural societies undergo profound transformations, the meaning and characteristics of rurality are continually redefined. Variations exist not only between different villages but also in how residents perceive rurality. Using social representation theory, the study of rurality has shifted focus from the physical space of rural areas to how rurality is understood, used, and symbolically represented through symbolic words [28,18]. M. Woods [39] believes that it is necessary to consider the political, economic, and social constructions that shape rurality.

Therefore, rurality is socially and culturally constructed. Linguistic representation is an important form of presentation. Baylina and Berg [40] found that images of rural idyll were constructed based on the themes of home, family, nature, and rural lifestyle. Phillips, Fish, and Agg [41] found that the gentry exhibit certain characteristics in the consumption of rural areas, prefer green living space, and follow a specific rural culture. Urban gentrification typically emphasizes bucolic landscapes – characterised by elements such as farmland, country lanes, and pastoral scenery – alongside leisure and entertainment infrastructures including restaurants, pubs, art galleries, museums, and sports facilities (Ramazannejad, Zarghamfard, Hajisharifi, and Azar, 2021). Gentrification also reflects different rural characteristics and preferences of local groups. Horton [42] noted that postmen became an important representation of rural idyllic landscapes in Britain. Further, McCormack [43] found that children's construction of rural space is based primarily on natural and agricultural elements. Children have different perceptions of rurality based on their own physical and discursive experiences. Zhou [44] observed that local culture significantly influenced the construction of rural tourism destination image; local and international idyllic rural images differed in certain ways. Moreover, rural areas are characterised by heterogeneity, dynamism, diversification and difference, and more attention should be paid to rural social and cultural space [45,46]. The social representation of space approach considers rural space as an ideal concept concerned with socio-cultural space, while the definition of villages based on statistical indicators considers rural space as physical space. Therefore, the focus is on constructed and deployed villages in different socio-cultural landscapes, rather than defining villages geographically using statistical indicators.

The construction of rurality leads to differing understandings of rural tourism sites depending on social backgrounds. Representations of rurality are dynamic and continually evolving. Social representation theory provides a unique and indispensable framework for understanding and predicting social phenomena, particularly as rural tourism undergoes significant change and societal perceptions of rural tourism evolve. Applying this theory to rurality development allows researchers to capture the shared understanding of rurality shaped by tourism and social culture, while also tracing the evolving process of rurality's social representation.

Despite its importance, few studies have focused on the representation of rurality and the evolution of rural tourism destinations, and theoretical frameworks explaining changes in individual cognition and differences in rurality are lacking. Given the diversity of rural attractions, a detailed classification of rurality representations is urgently needed. This study therefore applies social representation theory to examine rurality representations and their transformations, providing new insights into the study of rurality. The conceptual framework, based on social representation theory, is illustrated in Fig 2. In this figure, the surface of each triangle represents the understanding of rurality held by subjects 1 and 2 regarding

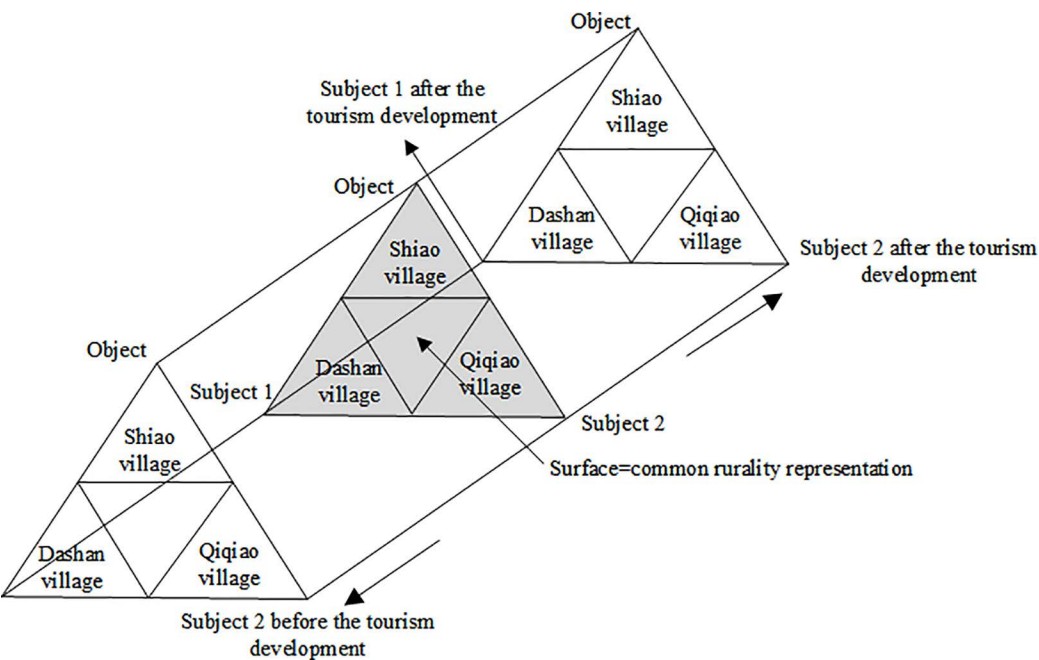

**Fig 2. Conceptual model of research.**

object o, capturing the diverse common senses prevailing at a given time. Fig 2 also depicts the dynamic process of social consensus in rurality representations across three rural tourism destinations.

## 3. Materials and methods

### 3.1 Research context

Nanjing's substantial economic foundation has supported the development of numerous rural tourism sites in its suburban areas. Studies indicate that rural tourism in Nanjing is primarily concentrated in Jiangning and Gaochun districts [47,48]. Accordingly, this study selected Shiao Village, Dashan Village, and Qiqiao Village as case sites: Shiao Village is located in Jiangning District, while Dashan and Qiqiao Villages are situated in Gaochun District.

To ensure a structurally representative sample of rural tourism destinations, this study focused on agritainments – operational farms that diversify into recreational, hospitality, and educational services to generate additional revenue. Shiao and Dashan Villages exemplify the agritainment model, offering activities such as farm stays and rural culinary experiences integrated into the farm environment. In contrast, Qiqiao Village represents traditional village tourism, emphasising the preservation and contextualisation of historical, architectural, and cultural heritage.

Shiao Village, located at the foot of the Niushou Mountain scenic area, is a typical homestay-focused catering tourism site. In 2012, the village completed renovations and environmental improvements, dedicating itself fully to tourism. Dashan Village, supported by the development of Gaochun International Cityslow beginning in 2010, evolved into a rural leisure tourism destination in the outer suburbs of Nanjing, offering primarily homemade food and accommodation services. Qiqiao Village is renowned for its traditional architecture from the Ming and Qing dynasties and, in 2013, was included in the catalog of traditional Chinese villages. It also serves as the largest gathering site for descendants of Confucius south of the Yangtze River.

All three case sites share exceptional natural settings, rich cultural heritage, and popularity as tourist destinations. However, Qiqiao Village exhibits less tourism development and fewer buildings, offering visitors primarily a local architectural

sightseeing experience. Shiao and Dashan Villages, by contrast, are characterised by large-scale tourism attractions, scenic landscapes, cultural cores, and catering services. Rural tourism development in these areas began predominantly around 2010. For analytical purposes, the year 2000 is designated as the pre-development period, and 2020 as the post-development period, providing a ten-year interval before and after tourism expansion to capture significant changes.

Since substantive rural tourism development in all three sites began around 2010, this year was used as a temporal pivot to examine the evolution of rurality representations. By extending ten years in both directions, the year 2000 and the year 2020 were established as critical reference points for interviews—representing the pre-development period and the post-development period, respectively. This framework enables a comparison of typical rurality representations and their transformations between the recalled past and the observed present. Interview data were collected during field surveys conducted from June 14–15, 2020 (Shiao Village) and August 27–30, 2020 (Dashan and Qiqiao Villages). These interviews captured contemporary representations of rurality alongside retrospective narratives of the pre-tourism era. By contrasting recalled past experiences with current realities, this methodological approach illuminates changes in the constitutive dimensions of rurality representations.

### 3.2 Methodology

Qualitative analysis allows for the generalisation and synthesis of dimensions and changes in the representation of rurality, drawing from the lived experiences and narratives of individuals in rural contexts [49]. Semi-structured interview transcripts were coded and analysed to identify themes and categories of rurality within rural tourism. An inductive, three-stage data analysis was conducted using a grounded theory approach. First, open coding was applied to conceptualise sentences and words within texts describing rural areas. Multiple levels of conceptualisation were identified at this stage, with some overlap between concepts; concepts appearing with low frequency (two instances or fewer) were excluded. Second, axial coding was performed to examine the internal logical relationships between concepts. This process refined category definitions, summarised connections among categories, and ultimately identified 28 primary categories. Finally, through selective coding, core categories were extracted from the primary categories, resulting in the identification of three overarching core categories of rurality.

### 3.3 Data collection

To comprehensively capture the representation and evolution of rurality, this study engaged two key stakeholder groups: local residents and tourists. In-depth interviews with residents provided a longitudinal perspective on changes in rurality, reflecting their direct lived experiences over time. Interviews with tourists, by contrast, captured the external perspective and immediate perceptions that influence rural consumption. This dual-perspective approach allows the social construction of rurality to be examined from both insider (resident) and outsider (tourist) viewpoints, offering a more holistic understanding.

Notably, a significant proportion of tourists were repeat visitors from nearby urban areas, enhancing both their familiarity with the destinations and the depth of their insights. Tourist interviews focused on perceptions of rurality following the development of tourism, recognising that rurality in these destinations is often constructed to serve visitors, and that its discursive expression is essential for understanding cognitive representations of rurality. The study also compared perceptions of rurality between resident and visitor groups, emphasising the role of both groups in shaping representations rather than relying solely on inter-site comparisons.

Interviews were conducted with fifty villagers and seventeen tourists on June 14–15 and August 27–30, 2020. Villagers are abbreviated as V, with interviews in Shiao, Dashan, and Qiqiao Villages labeled as VSA, VDS, and VQQ, respectively; tourists are similarly coded as TSA, TDS, and TQQ. During qualitative coding, no new core categories emerged after the 40th interview, indicating that data saturation had been achieved. The demographic characteristics of the interviewees are presented in Table 1.

**Table 1. Demographic characteristics of interviewees.**

| Item | type | amount | percent | item | type | amount | percent |
|---|---|---|---|---|---|---|---|
| gender | male | 31 | 46.27% | site | Shiao(SA) | 33 | 49.25% |
|  | female | 36 | 53.73% |  | Dashan(DS) | 26 | 38.81% |
| age | 18 and under | 3 | 4.48% |  | Qiqiao(QQ) | 8 | 11.94% |
|  | 19~30 | 7 | 10.45% | education | junior middle school or below | 31 | 46.27% |
|  | 31~45 | 27 | 40.30% |  | senior high school (including technical secondary school) | 18 | 26.87% |
|  | 46~60 | 22 | 32.84% |  | college | 15 | 22.39% |
|  | 61 and over | 8 | 11.93% |  | bachelor degree or above | 3 | 4.47% |
| role | tourist(T) | 17 | 25.37% | Residents participate in tourism or not | yes | 36 | 72.00% |
|  | villagers(V) | 50 | 74.63% |  | no | 14 | 28.00% |
| Residents' resource of income (multiple choice) | agriculture | 1 | 2.00% | tourist's occupation | company staff | 14 | 82.35% |
|  | hospitality industry | 33 | 66.00% |  | civil servants and public institutions | 1 | 5.88% |
|  | others (go out to work and work in local units) | 16 | 32.00% |  | self-employed | 2 | 11.76% |
| years of local residence | 10 years and below | 6 | 12.00% | Residents' income of family in a year | ten thousand and below | 4 | 8.00% |
|  | 10~20 years | 7 | 14.00% |  | 10~50 thousand | 10 | 20.00% |
|  | 20~30 years | 6 | 12.00% |  | 50~100 thousand | 17 | 34.00% |
|  | 30 years and above | 31 | 62.00% |  | 100 thousand and over | 19 | 38.00% |

## 4. Results

Through the three-stage grounded theory coding process, a hierarchical structure of rurality representation was developed, comprising three core categories, six main categories, and twenty-six subcategories. Fig 3 provides a comprehensive overview of both the dimensional structure and the evolutionary trajectory of rurality representation, which are described in detail in the following sections.

### 4.1 Material rurality representation

Material representation primarily encompasses the natural environment – such as agricultural and natural landscapes, animals, water, and environmental sanitation – and architectural elements, including settlements, architectural styles, rural roads, and leisure and infrastructure spaces. Table 2 presents the coding process for material rurality representation.

**4.1.1 Natural environment.** From the perspective of natural landscapes, before the development of tourism, rural scenery was primarily characterised by agricultural and natural elements. Specific representations included 'farmland', 'rape flowers', 'fields and gardens', 'bamboo', and 'vegetable gardens'. As VDS-3 recalled, 'In 2005, when you came, you would find that the biggest feature was that it was all fields; from one side to the other, there were nothing but fields'. VDS-10 similarly emphasised, 'This piece of land, before tourism development, was entirely fields. There were several vegetable plots in this area. Every household grew rice, vegetables, rapeseed, and other small crops'. Following the development of tourism, representations of rurality shifted from functional agricultural landscapes toward more ornamental features, such as 'peach blossom forests' and 'maple forests'. Tourism weakened the emphasis on agricultural production while enhancing the visibility of decorative natural elements. The diversity of elements characterising rurality expanded. For example, VDS-17 noted, 'There are canola flowers in the spring, maple leaves in the autumn, and around October 1, visitors can come and enjoy river crabs'.

Animals were also a strong component of rural representation, including 'fowl', 'hare', 'pheasant', 'eagle', and 'woodpecker'. VSA-20 described Shiao Village, saying, 'In the past, there were rice fields and natural landscapes, and many hares and pheasants. After development, they disappeared and cannot be seen. Previously, the mountains were wild and

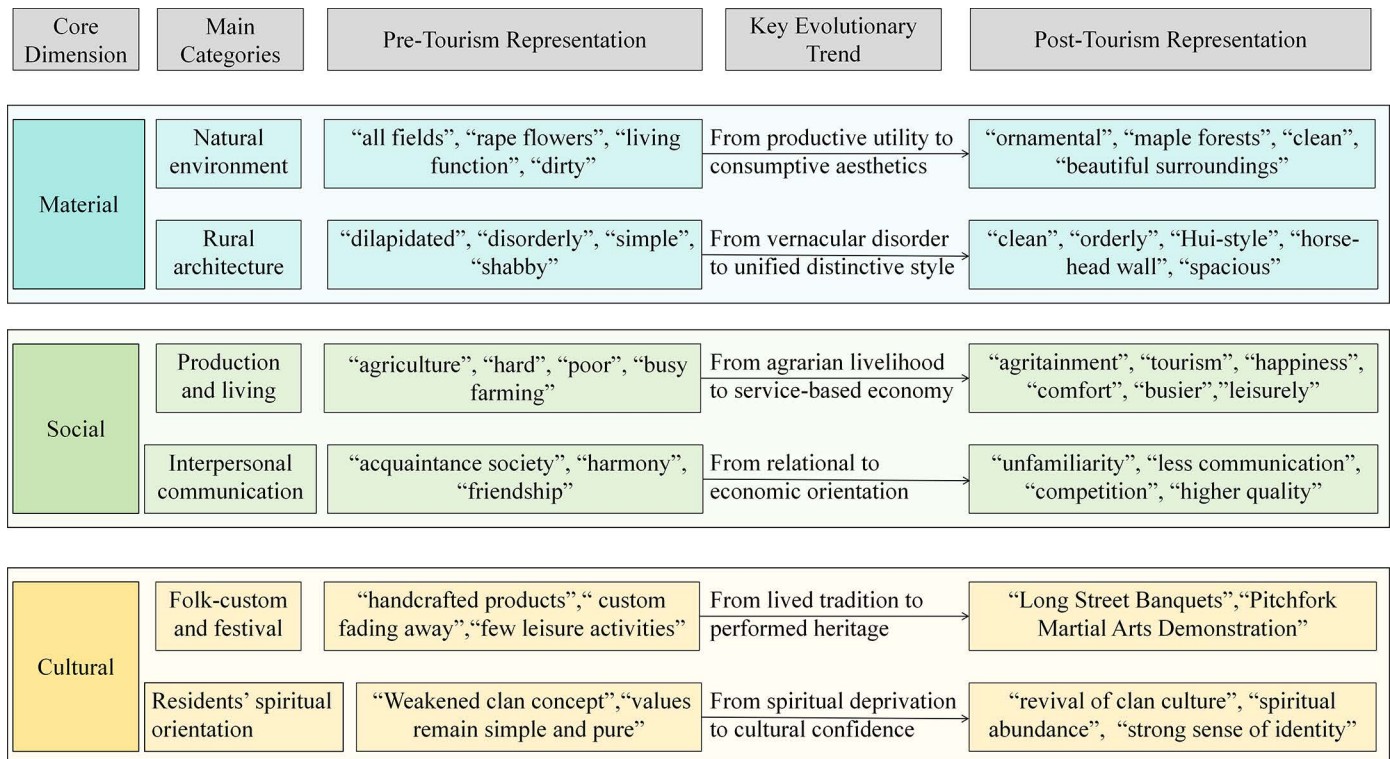

**Fig 3. The dimensional structure and evolutionary trajectory of rurality representation.**

**Table 2. The coding process of material rurality representation.**

| Selective coding | Axial coding | | Open coding | |
|---|---|---|---|---|
| Core category | Main category | Subcategory | Pre-Tourism development | Post-Tourism development |
| Material rurality representation | Natural environment | Natural scenery | Rice plant, Wheat, Rapeseed, Farmland, Vegetable plots, Bamboo | Peach blossoms, Lotus flowers, Maple leaves |
| | | Flora and fauna | Rapeseed flowers, Maple leaves, Eagles, Poultry, Hares, Pheasants, Woodpeckers, Cats/Dogs | Poultry, Cats/Dogs |
| | | Water bodies and resources | Washing clothes in the river, Washing rice in the river, Washing vegetables in the river, River water irrigation | Regularly cleaned, Ornamental |
| | | Climate and environment | Fresh air, Picturesque landscape (with hills and streams), Poor sanitation, Underdeveloped | Picturesque and tidy environment |
| | Rural architecture | Settlement landscape | Derelict, Decrepit, Chaotic | Tidy, Methodical |
| | | Transportation facilities | Muddy, Paths | Asphalt, Cement, Pavement, Spacious |
| | | Architectural features | Uniform, Shabby, Disordered, Ordinary | Black and white, Ancient dwellings, Hui-style, Horse-head gables, Comfortable |
| | | Leisure facilities | Benches at home entrance | Small long corridors, Pavilions |

hard to reach; now, with tourism, human traces are everywhere'. This illustrates a shift from agriculture-centred representations toward human-centred ones as tourism developed.

Water sources, which previously served practical functions such as 'washing rice', 'washing vegetables', 'washing clothes', and 'irrigation', have evolved toward ornamental and recreational purposes. In Shi'ao Village, ponds transitioned from functional uses to decorative features, whereas Dashan Village maintained both living and ornamental functions. VSA-19 explained, 'The pond water quality is good. We did not have running water when we were young, so we drank from ponds. The pond bottom was clear. Rice and vegetables were washed in the pond; now everyone uses tap water'.

Rural environments have also improved significantly. Residents' descriptions shifted from 'dirty' and 'backward' to expressions such as 'beautiful surroundings', 'cleanliness', and 'nice appearance'. JSA-14 recalled, 'When thinking of the countryside, the first thing that comes to mind is the dirt road, which is inconvenient. Every family raised stinky, dirty pigs or chickens. Now the environment is much better'. JSA-9 added, 'It's a little better than before; it's cleaner, the roads are tidy, and people are cleaning'.

Overall, residents' representations indicate a transformation from a focus on productive – landscapes – featuring functional farmland, wildlife, and utilitarian environments – to consumptive landscapes deliberately designed for aesthetic and recreational appeal. Ornamental features such as peach blossoms and maple forests exemplify this shift. This evolution reflects the social construction and ongoing negotiation of rurality, as theorised in social representation theory, and highlights a fundamental revaluation of nature – from a source of subsistence to a visual and recreational commodity shaped by tourist expectations.

Tourists' representations of rurality emphasise idyllic features, including fields, animals, vegetable plots, small ponds, flowers, grasses, serene environments, and fresh air. TDS-2 described a scene in a traditional village: 'A peasant leads a cow across a small bridge, with some grass houses nearby. I think this picture represents an ideal village, which is now impossible to see'. Comparative analysis reveals that residents perceive rurality through concrete, reality-based attributes, whereas tourists tend to idealise rural areas as aesthetically pleasing and socially constructed landscapes.

**4.1.2 Rural architecture.** The settlement landscape of rural tourism has undergone substantial transformation, reflected in the expansion of village areas, road construction and repair, improvements in architectural style, and the development of public recreational spaces. Tourism development has, to some extent, compressed natural rural spaces. Efforts to support tourism included repairing and planning the village settlement pattern, removing dilapidated structures such as farm walls, pigsties, toilets, and earthen constructions, and carrying out overall landscape maintenance. Consequently, rural settlement landscapes evolved from being 'dilapidated' and 'disorderly' to 'clean' and 'orderly'. VSA-12 noted, 'There are very big changes in the village. The current landscape is fully built and remodeled. It wasn't like this before – there was no such good scenery, no convenient traffic, and the houses were not so beautiful. Everything has been renovated'.

Architectural styles in rural areas have also changed significantly. Most rural houses in Nanjing were traditionally constructed with blue bricks and gray tiles, reflecting simplicity and elegance. Over time, building quality improved. The building conditions in rural areas are also gradually improving. The architectural style of rural areas evolved from the original 'simple' and 'shabby' to the present 'storied building' and 'comfortable' features. The dwellings were typical Double-storey bungalows. After tourism development, the Hui-style horse-head wall was added and the entire façade of the village was painted white to create a typical rural architectural style. The style of rural architecture evolved from originally 'dilapidated' and 'ordinary' to 'horse-head wall', 'Hui-style', and 'comfortable'. VSA-17 was asked if he liked the construction of Hui style architecture in the countryside. He said, 'It looks pretty good. It's definitely better than before'.

The representation of rural roads evolved from the original 'muddy roads' and 'small roads' to 'concrete', 'asphalt', 'spacious', and other rural infrastructure with modern characteristics. The overall representation of the village changed from the original special typical traditional village to a modern village. VSA-2 proposed changes in rural infrastructure that 'everything is very clean and greenery is good, while the previous rural areas were made of mud and no cement'.

In addition to the enhancement of the rurality in the architectural style, the leisure and infrastructure spaces expanded. Visitor centers, RV camps, pavilions, leisure corridors, and parking lots were constructed to accommodate tourists. Dashan Village, for example, established a slow city visitor center and RV holiday camp, while Qiqiao Village built a tourist reception center. VSA-15 observed, 'Tourists come to the recreation square to rest and sit under the pavilion'.

Residents' descriptions of rural architecture indicate that Huizhou-style vernacular features – such as horsehead gables, traditional design elements, cleanliness, and comfort – now characterise rural dwellings. The perceived rurality of architecture and infrastructure has shifted markedly, from 'untidy', 'dilapidated', 'messy', and 'ordinary' to 'neat' and 'spacious'. In these villages, built-up areas have expanded, with clear improvements in infrastructure, sanitation, housing quality, and architectural style. Tourism development has significantly enhanced the overall built landscape, elevating perceived rurality at the microscale, particularly from the perspective of tourists. This transformation demonstrates tourism's active role in reshaping rurality through architectural and settlement forms.

Tourists, in particular, expect rural buildings to exhibit distinctive local features with traditional vernacular styles. TDS-3 described a village in Jing County, Anhui Province: 'There are some well-preserved Hui-style buildings, all about 700 years old, with a small bridge and flowing water. The traditional village felt very authentic. Some places lacked certain features, but each had its own characteristics'. These observations illustrate that tourists and residents often have divergent expectations: residents value functional and aesthetic improvements, while tourists prioritise historically distinctive and visually appealing built environments that satisfy their desire for novelty, cultural engagement, and aesthetic pleasure.

## 4.2 Social rurality representation

Social representation encompasses the daily lives and activities of people, including modes of production, agricultural productivity, quality of life, pace of life, transportation, markets, diet, and local dialects, as well as interpersonal communication, such as neighborly relations and residents' social conduct. Table 3 outlines the coding process for social rurality representation.

**Table 3. The coding process of social rurality presentation.**

| Selective coding | Axial coding | | Open coding | |
|---|---|---|---|---|
| Core category | Main category | Subcategory | Pre-Tourism development | Post-Tourism development |
| Social rurality presentation | Production and living | Production methods | Farming | Farmhouse cuisine, Agritainment |
| | | Production tools | Manual labor, Animal power | Machinery |
| | | Labor and livelihood | farming, Toilsome, Poultry raising | Easier, Less free |
| | | Pace of life | Busy farming season, Leisurely | Leisurely, Busy |
| | | Information technology | – | QR code payments, Mobile phones |
| | | Transportation means | Bicycles | Buses, Private cars |
| | | Consumption patterns | Rarely going out | Express delivery |
| | | Local cuisine | Monotonous, Limited | Healthy, Western-style |
| | | Local language | Dialect | Mandarin |
| | Interpersonal communication | Neighborhood relations | Strong bonds, Mutual aid, Deep-rooted Ties | Competitive relations |
| | | Interaction principles | Relational exchanges | Transactional exchanges |
| | | Rural customs | Unspoiled and sincere, Industrious and frugal, Assiduous, Heartfelt hospitality | Unspoiled, Diligent |
| | | Resident quality | There are some illiterate individuals | Legal awareness, Educational attainment, Follow current affairs |

**4.2.1 Production and living.** In terms of modes of production, rural representation shows a clear shift from 'agriculture' to 'agritainment' and 'tourism', reflecting a transition from a primarily productive agricultural economy to a consumptive service sector. Shiao Village, for example, was historically focused on rice cultivation. At the turn of the century, some villagers had migrated for work. With the development of rural tourism, villagers returned to open farm-based restaurants and guesthouses, often collaborating with external investors. VSA-1 noted, 'Shiao Village is represented by homemade cooking'.

Agricultural production methods have also evolved, moving from labor- and animal-based practices to mechanisation. Traditional tools and draft animals have been replaced with machinery, improving efficiency. VSA-20 recalled, 'In the past, every household had cattle, sometimes shared among several families'.

Quality of life in rural areas has improved substantially. Villagers historically associated agricultural life with 'hardship' and 'poverty', whereas tourism development has introduced 'happiness' and 'comfort'. JAS-2 reflected on this change: 'Of course life is better now. Farming life in the past was exhausting and income was low. Now I can earn two or three thousand yuan per month, and I am happy'.

The pace of life has also shifted. Traditionally, villagers experienced periods of intense agricultural labor, but with rural tourism, life oscillates between 'busier' periods when guests are present and 'leisurely' times otherwise. VSA-19 described, 'Before tourism development, we were busy with farming during the season and worked outside at other times. Now we run an agritainment business. When guests are here, we cook and serve; when there are no guests, we rest'.

Transportation in rural tourism destinations has improved, moving from scooters, tractors, and bicycles to buses and private cars, reflecting a modernisation of rural infrastructure. Similarly, rural markets have expanded from small local markets to larger commercial hubs, benefiting from improved transportation and increased visitor demand.

Dietary practices have diversified as well. Traditional subsistence diets have given way to richer, more nutritious offerings. The 'homemade cook' has become a prominent feature of rural tourism, with local dishes such as hen soup and river crab serving as important cultural representations.

Language practices have also been influenced by tourism. Dashan and Qiqiao Villages have unique regional dialects, with Dashan falling within the Wu language area. Tourism has increased residents' use of Mandarin, impacting local language habits. VDS-01 observed, 'Before tourism developed, everyone spoke dialects. Now they all speak a little Mandarin'.

Overall, the mode of production has undergone a profound transformation, shifting from traditional agriculture to diversified industries like farm stays and rural tourism. This has enhanced the quality of rural life, moving from hardship and poverty to happiness and contentment. Daily life, including aspects such as poultry farming, diet, shopping, and transportation, has modernised, and Mandarin proficiency has increased, reflecting broader social and technological modernisation. Compared with traditional rural life reveals a certain degree of attenuation of rurality, with rural life becoming more modernized. The development of social productive forces has had a profound impact on the traditional rural lifestyle [50]. Consequently, the manifestation of rurality in daily life exhibits a strong characteristic of progressiveness, aligning with contemporary trends.

Tourists' perceptions of rural production and landscapes often emphasise idealised imagery. Typical representations include 'smoke rising from kitchen chimneys', 'enjoying the cool', 'playing chess or cards', and 'wood-burning stoves'. TDS-2 remarked, 'I need to use this wood stove to cook, and when I visit a rural village, I must first catch a chicken'. TDS-4 described a village scene: 'Smoky air, children playing, summer evening atmosphere, and the sense of enjoying the cool after dinner'. Additionally, TDS-2 noted that early-morning pond scenes, with residents washing clothes using wooden implements, are still preserved in Dashan Village. These descriptions show that tourists often idealise traditional rural life, while residents seek convenience and modernisation. This contrast illustrates that tourists' representations of rurality are shaped by idealised, socially constructed expectations of the rural experience.

**4.2.2 Interpersonal communication.** Neighbourly relations in rural areas have undergone a marked transformation, shifting from the traditional 'acquaintance society' characterised by 'harmony' and 'friendship' to a present state often

described as 'unfamiliarity', 'less communication', and 'competition'. Urbanisation has prompted many villagers to migrate or adopt urban lifestyles. Concurrently, tourism development has attracted external investors, disrupting local social networks and prompting new interactions among residents. As the tourism industry became a primary source of income, communication among villagers declined, and competitive dynamics emerged. VDS-11 observed, 'Some people feel psychologically unbalanced when they see others' business doing well. People's hearts are not as pure as they used to be. There may be some competition with each other'.

The overall quality of life of residents has improved significantly. Social representations of villagers have evolved from being relatively low in education and civic awareness to higher-quality, more informed standards. JDS-11 noted, 'At that time, there were many illiterates and weak legal awareness. Now that my educational level has improved, I am more concerned about current affairs and other issues'. In terms of interpersonal communication, rurality representation has generally weakened, particularly in areas with high levels of urbanisation and tourism development. Resident relocation and the influx of external businesses have altered the original social atmosphere, while market-oriented values have reshaped perceptions and social norms.

Overall, the social fabric has shifted from traditional values such as 'loyalty', 'mutual assistance', 'simplicity', 'cooperation', and 'low cultural attainment' toward 'competition' and 'higher cultural quality'. This erosion of the traditional 'acquaintance society' reflects a fundamental social restructuring, marking a transition from peasant-economy-based interpersonal relationships to market-driven, interest-oriented relations. The solidarity once rooted in shared agricultural livelihoods and kinship ties has been supplanted by impersonal, contract-based interactions characteristic of a market economy. Tourism has acted as a catalyst for this transformation, with emerging competition among villagers representing a tangible manifestation of this new economic rationality, illustrating the social costs embedded within tourism's economic benefits.

From the perspective of tourists, the simple and genuine social interactions of rural residents remain a key aspect of rurality and an important distinction from urban life. Villagers, as the primary agents shaping rurality, provide an interpersonal experience that is central to the appeal of rural tourism. While tourists often recognise this, local residents – motivated by immediate economic gains – may not fully appreciate their role in constructing this representation. Tourism development has occasionally disrupted these interactions, yet elements of traditional social cohesion persist. YDS-2 commented, 'In fact, after the development of tourism here, it is a bit materialised and commercialised, but the customs and morals of the people are still very pure. This place was not as commercialised as other places after tourism development. I think the people here are nice'.

### 4.3 Cultural rurality representation

Cultural representation reflects the subjective and spiritual dimensions of rural life, encompassing festival activities – such as folk customs, traditional celebrations, leisure, and entertainment – and spiritual orientations, including clan culture, historical relics, residents' values, and sense of identity. Table 4 presents the coding process for cultural rurality representations.

**4.3.1 Folk-custom and festival.** Cultural representation reflects the subjective and spiritual dimensions of rural life, encompassing festival activities – such as folk customs, traditional celebrations, leisure, and entertainment – and spiritual orientations, including clan culture, historical relics, residents' values, and sense of identity. Table 4 presents the coding process for cultural rurality representations [51]. VDS-5, a resident of Dashan Village, stated, 'The Tiandi Stage has always existed, but it may have been improved after tourism development. The Tiandi Stage has a history of 3,800 years. Every year, on the 17th, 18th, and 19th of March in the lunar calendar, some local Huangmei opera or Yue opera troupes from Anhui are invited to perform for three to four days'.

The festival representation in these villages largely aligns with general rural areas, including traditional festivals such as the Spring Festival and Dragon Boat Festival, which have been adapted into events like the Crab Festival and Golden Flower Festival, reflecting tourism influences. Traditional customs such as making zongzi, writing and pasting couplets,

Table 4. The coding process of cultural rurality representation.

| Selective coding | Axial coding | | Open coding | |
|---|---|---|---|---|
| Core category | Main category | Subcategory | Pre-Tourism development | Post-Tourism development |
| Cultural rurality representation | Folk-custom and festival | Rural folklore | Lost | Horse-Lantern Dance, Opera Performances, Long-Street Banquets, Pitchfork Martial Arts Demonstration, Jinhua Festival |
| | | Traditional festivals | Making Zongzi, Writing couplets, Pasting couplets | Consumption-oriented |
| | | Recreation and leisure | Lacking | Square Dancing, Strolling, Guandan (A popular chinese card game) |
| | Residents' spiritual orientation | Clan and ancestral halls | Disregarded | Genealogies, Cultural preservation, Inheritance |
| | | Spiritual beliefs | Baoping well, Wenfeng pagoda, Niushou mountain, Pagoda, Buddhist relics | Baoping Well, Wenfeng Pagoda, Niushou Mountain, Pagoda, Buddhist Relics |
| | | Values and outlook | Diligence, Striving for improvement | Better living, Positive, Enjoyment of life |
| | | Local identity | Rural poverty | Rural comfort |

dragon and lantern dances, performing operas, Long Street Banquets, Pitchfork Martial Arts Demonstrations, square dancing, casual walks, and playing Guandan have been revitalised. Tourism has facilitated the reproduction of these local festivals, making them significant attractions and drivers of rural tourism. VDS-7 remarked, 'For example, we performed the 'Five Fierce Deities' (Wuchang Dance), a traditional folk dance to drive away evil spirits and pray for peace. We had neglected it for a while, but now, with tourism development, it has regained importance as a key cultural representation'.

In terms of leisure and entertainment, traditional villages offered few activities, whereas modern rural tourism destinations now provide a rich and varied array of recreational options, including public square dancing and the card game 'Guandan'. Folk costumes and festival activities in rural destinations have undergone notable transformations, reflecting the revival of vibrant rural folk culture. Examples include the Long Street Banquet – a communal feast symbolising unity and prosperity – the Five Fierce Deities Dance, rooted in folk beliefs for warding off evil, and the Pitchfork Martial Arts Demonstration, a heritage performance. Tourism has enhanced the display of these activities, contributing to the cultural renaissance of rural areas.

Tourists also show interest in rural customs, though their perceptions tend to emphasise typical and widely recognised elements such as 'lively' atmospheres, dragon lantern performances, and lion dances. TDS-4 noted, 'There are still dragon and lion dances in every rural area of China. When festivals arrive, these continue. You can witness them when you visit. They are lively, though somewhat less so than in the past'. Tourists' representations generally reflect national or mainstream cultural practices, whereas residents highlight region-specific activities. This distinction underscores the inherent regional variability in the concept of rurality.

**4.3.2 Residents' spiritual orientation.** The concept of clans in villages has gradually weakened in recent years. Shiao Village, a natural village inhabited primarily by the Tang clan, retains a general awareness of clan origins, but lacks an ancestral hall and hosts few traditional clan sacrificial activities, leading to a dilution of clan culture. Dashan Village, inhabited by the Rui clan, maintains a Rui clan ancestral hall with preserved ancestral tablets, while Qiqiao Village, home to descendants of Confucius, has a newly constructed Kong family ancestral hall. Historically, family structures, genealogies, and ancestral instructions shaped rural life, but modernisation has led to the erosion of traditional clan culture. VDS-2 noted, 'No one paid attention to it when we were young, but now residents have begun to have awareness of repairing genealogy, and it is necessary to protect culture'.

Historical relics represent another dimension of rural spiritual culture. Shiao Village, situated at the foot of Niushou Mountain, preserves the site of Yue Fei's anti-Jin battle, including trenches and ruins from the historic Niutoushan victory. Tourism development has amplified rural spiritual culture, reinforcing local heritage through initiatives such as constructing the 'Sanbao Road' and naming the reservoir near Zheng He's tomb as 'Zheng He Lake'. In Dashan Village, the Ming-era Wenfeng Pagoda, destroyed during the Anti-Japanese War, was restored as part of tourism-led heritage conservation efforts.

Traditional rural values – harmony, filial piety, frugality, hard work, kindness, and integrity – have been partially eroded by modern economic imperatives, shifting toward more 'open' and market-oriented perspectives. Nonetheless, virtues such as simplicity, diligence, thrift, and filial piety remain prevalent, and villagers' quality and cultural literacy have improved. Rural economic transitions, from smallholder subsistence to market-oriented activity, have fostered more rational economic behaviours while maintaining cooperative social norms. JSA-21 commented, 'Every family comes here to help if there is something in the family. This kind of thing has continued. As before, people can do this automatically without saying anything'.

Tourism has also strengthened residents' sense of identity and attachment to their villages. Previously, rural areas were under-recognized due to poverty, but tourism has enhanced both material benefits and cultural self-confidence. Representations include 'like the rural', 'accustomed to the rural', and 'identified with the rural'. VDS-8 stated, 'The development of tourism should have an impact on the identity and cultural self-confidence of farmers, and make them more self-confident'.

For residents, key spiritual and cultural aspects of rurality include clan culture, the Yue Fei Anti-Gold Fortress, Wenfeng Pagoda, rational economic orientation, and cultural confidence. Tourism has facilitated the excavation, dissemination, and promotion of these elements, enabling residents to perceive and value the cultural and spiritual heritage of their villages more vividly. For tourists, the spiritual dimension of rurality is experienced differently, focusing on a 'quiet atmosphere', 'simple folk customs', and 'spiritual abundance'. In Dashan, as part of the slow international city initiative, the slow pace of life is a core representation of rurality. YDS-2 remarked, 'When you go to this place, you basically do not need to think too much about it. You can integrate yourself into such an environment; you will be very relaxed, very comfortable, and very at ease, especially in a quiet life'. Tourists often yearn for these aspects of rural living, which are perceived as positive contrasts to the fast-paced urban environment, making spiritual and cultural elements central to rural tourism appeal.

## 5. Conclusions and discussion

From the perspective of social representation theory, the selection of Shiao, Dashan, and Qiqiao villages as case studies provides a comprehensive summary of people's cognition of rurality. This research synthesises the hierarchical structure and transformations of rurality representation, highlighting both residents' and tourists' typical images of rural areas.

Firstly, this research confirms the basic dimensional structure of rurality representation. Although rural areas hugely differ across different countries and regions, and the connotation of rural areas develops and changes with the social background, the basic representational elements can be clarified from the unified representation dimension of rurality, which is easy to compare. After the qualitative analysis of texts from semi-structured interviews on rurality, the main axis coding revealed three core categories: rural material, social, and cultural dimensions. Six main categories were then extracted. The study further corroborates existing research by Hu, Li, Zhang, Chen, and Yuan [52], who proposed the three-dimensional spatial structure [6].Rural areas have more extensive natural environments, harmonious interpersonal relations, and pleasant spiritual lives than cities [31,53]; indeed, these are the core elements of rurality. A measurement scale for rurality can be developed from the dimensions mentioned above.In addition, the representation of rurality in rural tourism destinations has changed from agricultural- to tourism-based in line with the transformation of rural society. Before the development of tourism, rurality was closely linked to agriculture, which was an important form of production and life in rural areas. However, after the development of tourism, rural tourism related elements became the central manifestation of rurality. The representation of agriculture weakened and that of tourism strengthened. As indicated by Fleischer and

Tchetchik [54] the development of rural has combined the heritage and culture to rural, brings new value to the farmers and turning the farmers into 'neo-peasants'.

Finally, residents and tourists have different cognitions on the connotation of rural representation. Based on the theory of social representation, different social groups form typical cognitive differences in rurality. The rural image described by tourists is an ideal rural scene, with birds and flowers, traditional architecture, interpersonal friendship, and spiritual wealth [55]. To some extent, it conceals social reality and represents a typical image of traditional rural area. The description of rural people is more realistic and reflects the inadequacy of rural development, with inconvenient transport and economic backwardness as basic features; residents look forward to a more practical and comfortable modern village. Tourists, by contrast, look for beautiful natural landscapes, differentiated cultures and the simple folk customs of rural people. They pay more attention to the elements missing in the cities and those constructed by tourists as idylls. The rural social construction of different groups depends on the interests of the group, from a theoretical perspective of the stakeholders [56].

## 6. Theoretical contributions

Our analysis demonstrates that the social representation theory provides a useful theoretical perspective to show the changes of common sense of rurality representation in rural tourism destination. Further, this perspective helps reveal common ideas about the rurality representation in a rural tourism destination. Changes in time and social groups have contributed to cognitive differences in rurality representations. Based on the social representation theory, different social backgrounds form different ideas about rurality [57]. Residents focus more on practical rurality representation and tourists focus more on idyllic landscapes representation. This study also confirms this finding. The apply of the social representation theory can support that the construction of rurality is social based and shapable.

This study affirms the need to consider the rurality as a pluralistic and subjective socio-cultural construct. The representation of the rurality produces different cognitions in local cultures, social backgrounds and groups, consistent with Sharpley's [58] assertion that the rurality representation is dynamic and determined by social interaction. From the perspective of social construction, Mormont [59] pointed out that rurality is a collection of concepts that implies a discourse describing the rural territorial unit and society. The connotation of rurality changes over time, and numerous modern rural tourism destinations moved away from agricultural features [60]. The emergence of a new rurality under the influence of tourism needs to be closely observed. This new rurality needs to consider the needs of tourists and the demands of residents. Under the guidance of the social representation theory, rurality must be represented in a manner that adapts to tourist destinations in the modern era. This new rurality emphasizes the construction of representations of rurality at the material, social and cultural dimensions.

## 7. Practical implications

Rurality is widely recognised as the foundation of rural tourism, underscoring its critical importance in shaping tourist destinations. Scenic areas should deliberately cultivate a perception of rurality that aligns with local culture while meeting the expectations of contemporary tourists. Each rural tourism destination should develop its own distinctive rural characteristics, drawing on its unique natural environment and cultural heritage. The representation of rurality must be grounded in these local attributes [61,62]. Different groups have different concerns about rurality. Rural tourism destinations further cater to major segmented market to create a high rurality perception. The analysis of the dimensional structure of rurality indicates that its construction should strategically target material, social, and cultural levels to strengthen the destination's competitiveness [63]. The scientific construction of rurality can weaken the phenomenon of homogeneity among destinations, promote the preservation and transmission of rural culture, and enhance tourist satisfaction, thereby contributing to the revitalisation of the rural tourism industry.

## 8. Limitations and prospects

This study has several limitations. First, it focuses on three typical rural tourism destinations in large cities, divided into two main categories. The dynamics observed in other types of rural tourism destinations may differ, and future research should undertake comparative studies across a broader range of rural tourism types. Given the diversity and heterogeneity of rural areas, caution is warranted when generalising these findings. Second, social representation theory primarily addresses the formation of social consensus. Future research could focus on constructing typologies of rurality representations specific to different rural tourism contexts, enhancing the practical application of the theory. Finally, significant variations in rurality representations were observed across different regional cultures at the case sites, indicating that cultural context plays a crucial role in shaping perceptions of rurality and should be carefully considered in subsequent studies.

## Supporting information

**S1 File. Interview recordings and translated transcripts from Shiao Village, Dashan Village, and Qiqiao Village.**
(ZIP)

## Acknowledgments

The author would like to express her deepest gratitude to all participants, for the valuable contribution to this study.

## Author contributions

**Data curation:** Yechen Zhang.

**Investigation:** Xinyi Li.

**Writing – original draft:** Xueting Hong.

**Writing – review & editing:** Zhenfang Huang.

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
