## [Decision Letter · Decision Letter 0]

5 Sep 2025

Dear Dr. Zhang,

Thank you for submitting your manuscript to PLOS ONE. After careful consideration, we feel that it has merit but does not fully meet PLOS ONE’s publication criteria as it currently stands. Therefore, we invite you to submit a revised version of the manuscript that addresses the points raised during the review process.

**I would recommend a major revision for this manuscript. Please revise that according the following comments  from reviewers.**

We look forward to receiving your revised manuscript.

Kind regards,

Qianda Zhuang, Ph.D.

Guest Editor

PLOS ONE

**Journal Requirements:**

1. When submitting your revision, we need you to address these additional requirements. Please ensure that your manuscript meets PLOS ONE's style requirements, including those for file naming. The PLOS ONE style templates can be found at https://journals.plos.org/plosone/s/file?id=wjVg/PLOSOne_formatting_sample_main_body.pdf and https://journals.plos.org/plosone/s/file?id=ba62/PLOSOne_formatting_sample_title_authors_affiliations.pdf 2. Please provide additional details regarding participant consent. In the ethics statement in the Methods and online submission information, please ensure that you have specified (a) whether consent was informed and (b) what type you obtained (for instance, written or verbal, and if verbal, how it was documented and witnessed). If your study included minors, state whether you obtained consent from parents or guardians. If the need for consent was waived by the ethics committee, please include this information. If you are reporting a retrospective study of medical records or archived samples, please ensure that you have discussed whether all data were fully anonymized before you accessed them and/or whether the IRB or ethics committee waived the requirement for informed consent. If patients provided informed written consent to have data from their medical records used in research, please include this information. 3. Thank you for stating in your Funding Statement: This work was supported by the National Natural Science Foundation of China [grant numbers 42301272]; and the Humanities and Social Science Foundation of Ministry of Education [grant number 22YJC790041].  Please provide an amended statement that declares *all* the funding or sources of support (whether external or internal to your organization) received during this study, as detailed online in our guide for authors at http://journals.plos.org/plosone/s/submit-now.  Please also include the statement “There was no additional external funding received for this study.” in your updated Funding Statement. Please include your amended Funding Statement within your cover letter. We will change the online submission form on your behalf. 4. We note that your Data Availability Statement is currently as follows: All relevant data are within the manuscript and its Supporting Information files. Please confirm at this time whether or not your submission contains all raw data required to replicate the results of your study. Authors must share the “minimal data set” for their submission. PLOS defines the minimal data set to consist of the data required to replicate all study findings reported in the article, as well as related metadata and methods (https://journals.plos.org/plosone/s/data-availability#loc-minimal-data-set-definition). For example, authors should submit the following data: - The values behind the means, standard deviations and other measures reported;- The values used to build graphs;- The points extracted from images for analysis. Authors do not need to submit their entire data set if only a portion of the data was used in the reported study. If your submission does not contain these data, please either upload them as Supporting Information files or deposit them to a stable, public repository and provide us with the relevant URLs, DOIs, or accession numbers. For a list of recommended repositories, please see https://journals.plos.org/plosone/s/recommended-repositories. If there are ethical or legal restrictions on sharing a de-identified data set, please explain them in detail (e.g., data contain potentially sensitive information, data are owned by a third-party organization, etc.) and who has imposed them (e.g., an ethics committee). Please also provide contact information for a data access committee, ethics committee, or other institutional body to which data requests may be sent. If data are owned by a third party, please indicate how others may request data access. 5. PLOS requires an ORCID iD for the corresponding author in Editorial Manager on papers submitted after December 6th, 2016. Please ensure that you have an ORCID iD and that it is validated in Editorial Manager. To do this, go to ‘Update my Information’ (in the upper left-hand corner of the main menu), and click on the Fetch/Validate link next to the ORCID field. This will take you to the ORCID site and allow you to create a new iD or authenticate a pre-existing iD in Editorial Manager. 6. Please amend either the abstract on the online submission form (via Edit Submission) or the abstract in the manuscript so that they are identical. 7. If the reviewer comments include a recommendation to cite specific previously published works, please review and evaluate these publications to determine whether they are relevant and should be cited. There is no requirement to cite these works unless the editor has indicated otherwise. 

Reviewers' comments:

**Comments to the Author**

1. Is the manuscript technically sound, and do the data support the conclusions?

Reviewer #1: Yes

Reviewer #2: Yes

2. Has the statistical analysis been performed appropriately and rigorously?

Reviewer #1: N/A

Reviewer #2: Yes

3. Have the authors made all data underlying the findings in their manuscript fully available?

Reviewer #1: No

Reviewer #2: Yes

4. Is the manuscript presented in an intelligible fashion and written in standard English?

Reviewer #1: Yes

Reviewer #2: Yes

**Reviewer #1:** Dear Author, Dear Author, Dear Author, Dear Author,

The aim of the paper, as presented in the title and abstract, is very clear. The abstract effectively outlines the purpose of the study.

The statement on page 4, lines 16 - 17 (“little is known about the content and change of rurality in rural tourism destinations”) should be substantiated and further explained. There appears to be a substantial body of literature on this topic; therefore, clarifying the specific perspective or contribution of the current article would strengthen its position.

Several statements throughout the manuscript require clarification, as they appear ambiguous or potentially affected by issues in English translation. The following examples illustrate this concern:

• “The rurality is the key attraction of tourism” - This claim seems overly generalized, as tourism encompasses a wide range of forms and motivations. A more nuanced formulation would be advisable to reflect the diversity of tourism drivers.

• “Given the diversity of quantitative research on rural representation of rurality as rural is diversity, there is a need to identify the sub-dimensions or components of rurality representation”.

• “sheep” (line 22, page 9.

The manuscript states that “This study focuses on the two periods before and after the development of tourism to see how rurality representations altered after social change”; however, the only temporal reference provided is “June 14–15 and August 27–30, 2020.” Greater clarity is needed. Furthermore, the statement on page 7 — “Rurality is not only a subjective perception but also a product of social construction” — is particularly significant. This claim would benefit from further elaboration and the inclusion of relevant scholarly references to strengthen its theoretical grounding and enhance its impact.

It can be inferred that “agritainment” derives from a combination of “agriculture” and “entertainment”; however, the term should be clearly defined, and it should be clarified whether it has been used in previous academic literature. The term “angertainment” (page 22, line 5) appears to be a likely misspelling of “agritainment.”

In the Methodology section, it is recommended to provide a more detailed explanation of the process through which the three core categories were distilled from the initial 28 main categories.

The opening lines of the Data Collection section should be clarified or reformulated, as they currently suggest that interviews with tourists were a plan B option due to an insufficient number of interviews with residents.

**Reviewer #2:** The issue of rurality is a crucial topic in rural tourism. This manuscript has important theoretical and practical value, and it is recommended to revise it before acceptance. Following are my observations: The issue of rurality is a crucial topic in rural tourism. This manuscript has important theoretical and practical value, and it is recommended to revise it before acceptance. Following are my observations: The issue of rurality is a crucial topic in rural tourism. This manuscript has important theoretical and practical value, and it is recommended to revise it before acceptance. Following are my observations: The issue of rurality is a crucial topic in rural tourism. This manuscript has important theoretical and practical value, and it is recommended to revise it before acceptance. Following are my observations:

In terms of the results report, the author presents the representation of rurality and its change in rural tourism destinations based on the Social Representation Theory. However, the authors did not strictly follow the procedures of the Grounded Theory analysis paradigm to present the outcomes of open coding, axial coding, and selective coding. Also, the results are superficial and descriptive, without in-depth analysis and

synthesis. The presentation of the results can be enhanced by including visual representations such as graphs to summarise key points and ensuring that the analysis goes beyond descriptive findings to provide in depth insights and synthesis. Suggest making appropriate modifications.

The third conclusion presents the differences in the perception of rurality between tourists and residents. However, the author does not strictly distinguish between tourists and residents in the result analysis section, leading to insufficient evidential support. Suggest making appropriate modifications.

The article still has some shortcomings in terms of writing and formatting. It is recommended that the authors carefully revise and refine the work. Also, it is suggested that the authors provide explanations for some rural customary activities mentioned in the results section, such as five rampages, guandan game, Long Street Banquet and so on.

Suggest supplementing relevant literature appropriately.

TANG Chengcai, LIU Yaru, WAN Ziwei, LIANG Wenqi. Evaluation system and influencing paths of the integration of culture and tourism of traditional villages. Journal of Geographical Sciences, 2023, 33(12): 2489-2510.

TANG Chengcai, Yang Yuanyuan, LIU Yaru, Xiao Xiaoyue. Comprehensive Evaluation of the Cultural Inheritance Level of Tourism-oriented Traditional Villages: The Example of Beijing. Tourism Management Perspectives, 2023, 48, 101166.

Tang Chengcai, Han Ying, Pin Ng. Green consumption intention and behavior of tourists in urban and rural destinations. Journal of Environmental Planning and Management, 2023, 66(10): 2126-2150.

Liu Yaru, Tang Chengcai*, Wan Ziwei. Multi-scenario analysis and the construction of the revitalization model of green development in tourism traditional villages. Journal of Resources and Ecology, 2023, 14(2): 239-251.

Chengcai Tang, Qianqian Zheng, Pin Ng. A Study on the Coordinative Green Development of Tourist Experience and Commercialization of Tourism at Cultural Heritage Sites. Sustainability,2019, 11, 4732.

Zhang Y, Zheng Q, Tang C, et al. Spatial characteristics and restructuring model of the agro-cultural heritage site in the context of culture and tourism integration[J]. Heliyon, 2024, 10(9).https://doi.org/10.1016/j.heliyon.2024.e30227.

.

Reviewer #1: No

Reviewer #2: No

---

## [Author Response · Author response to Decision Letter 1]

17 Nov 2025

We would like to express our sincere appreciation to the editors and reviewers for their time and invaluable suggestions, which have significantly enhanced the quality of our paper. In response to all the points raised, we have undertaken comprehensive revisions to the manuscript.

---

## [Decision Letter · Decision Letter 1]

28 Jan 2026

Dear Dr. Zhang,

Thank you for submitting your manuscript to PLOS ONE. After careful consideration, we feel that it has merit but does not fully meet PLOS ONE’s publication criteria as it currently stands. Therefore, we invite you to submit a revised version of the manuscript that addresses the points raised during the review process.

We look forward to receiving your revised manuscript.

Kind regards,

Qianda Zhuang, Ph.D.

Guest Editor

PLOS One

Journal Requirements:

Additional Editor Comments:

I believe the manuscript has improved a lot after the revision according to reviewers' comments. However, the language editing and polish is still required to meet the publication standards. Therefore  I recommend a thorough review and English editing of the manuscript before acceptance.

Reviewer's Responses to Questions

**Comments to the Author**

Reviewer #1: All comments have been addressed

Reviewer #3: All comments have been addressed

2. Is the manuscript technically sound, and do the data support the conclusions?

Reviewer #1: Partly

Reviewer #3: Yes

3. Has the statistical analysis been performed appropriately and rigorously?

Reviewer #1: (No Response)

Reviewer #3: Yes

4. Have the authors made all data underlying the findings in their manuscript fully available?

Reviewer #1: Yes

Reviewer #3: Yes

5. Is the manuscript presented in an intelligible fashion and written in standard English?

Reviewer #1: No

Reviewer #3: Yes

Reviewer #1: Dear Author,

The observations from the first phase of the review have been addressed. By also responding to the comments of the other reviewer, the paper now appears more solid.

However, the paper still needs more careful English editing. Here is just one example: “pre-development period (circa 2000) and the post-development era (circa 2020).” Since the exact dates of data collection are provided later in the text, the formulations “circa 2000” and “circa 2020” are not the most appropriate choices.

Reviewer #3: Dear Authors,

Thank you for carefully addressing and implementing all comments and suggestions provided in my review. The revised manuscript demonstrates a clear improvement in terms of clarity, structure, and methodological consistency.

The revisions have strengthened the overall quality of the paper, and the arguments are now more coherent and convincingly supported by the analysis. In its current form, the manuscript meets the expected academic standards.

Accordingly, I consider the manuscript suitable for publication and recommend it for acceptance.

Kind regards,

.

Reviewer #1: No

Reviewer #3: No

You may also use PLOS’s free figure tool, NAAS, to help you prepare publication quality figures: https://journals.plos.org/plosone/s/figures#loc-tools-for-figure-preparation

---

## [Author Response · Author response to Decision Letter 2]

14 Mar 2026

Dear Editor and Reviewers,

Thank you for giving us the opportunity to revise our manuscript. We have carefully considered all the comments and have prepared a point-by-point response to the reviewers' concerns. The detailed responses are listed in the attached document titled "Response to Reviewers."Should any further adjustments be required, we remain open to additional refinements.

Sincerely,

The Authors

---

## [Editor Report · Decision Letter 2]

30 Mar 2026

Rurality Representation and Changes in Rural Tourism Destination

PONE-D-25-26199R2

Dear Dr. Zhang,

We’re pleased to inform you that your manuscript has been judged scientifically suitable for publication and will be formally accepted for publication once it meets all outstanding technical requirements.

Kind regards,

Qianda Zhuang, Ph.D.

Guest Editor

PLOS One
---

## [Editor Report · Acceptance letter]

PONE-D-25-26199R2

PLOS One

Dear Dr. Zhang,

I'm pleased to inform you that your manuscript has been deemed suitable for publication in PLOS One. Congratulations! Your manuscript is now being handed over to our production team.

Kind regards,

on behalf of

Dr. Qianda Zhuang

Guest Editor

PLOS One